# DockingApp RF: A State-of-the-Art Novel Scoring Function for Molecular Docking in a User-Friendly Interface to AutoDock Vina

**DOI:** 10.3390/ijms21249548

**Published:** 2020-12-15

**Authors:** Gabriele Macari, Daniele Toti, Andrea Pasquadibisceglie, Fabio Polticelli

**Affiliations:** 1Department of Sciences, Roma Tre University, 00146 Rome, Italy; gabriele.macari@uniroma3.it (G.M.); andrea.pasquadibisceglie@uniroma3.it (A.P.); 2Faculty of Mathematical, Physical and Natural Sciences, Catholic University of the Sacred Heart, 25121 Brescia, Italy; daniele.toti@unicatt.it; 3National Institute of Nuclear Physics, Roma Tre Section, 00146 Rome, Italy

**Keywords:** docking, scoring, function, machine learning, random forest

## Abstract

Motivation: Bringing a new drug to the market is expensive and time-consuming. To cut the costs and time, computer-aided drug design (CADD) approaches have been increasingly included in the drug discovery pipeline. However, despite traditional docking tools show a good conformational space sampling ability, they are still unable to produce accurate binding affinity predictions. This work presents a novel scoring function for molecular docking seamlessly integrated into DockingApp, a user-friendly graphical interface for AutoDock Vina. The proposed function is based on a random forest model and a selection of specific features to overcome the existing limits of Vina’s original scoring mechanism. A novel version of DockingApp, named DockingApp RF, has been developed to host the proposed scoring function and to automatize the rescoring procedure of the output of AutoDock Vina, even to nonexpert users. Results: By coupling intermolecular interaction, solvent accessible surface area features and Vina’s energy terms, DockingApp RF’s new scoring function is able to improve the binding affinity prediction of AutoDock Vina. Furthermore, comparison tests carried out on the CASF-2013 and CASF-2016 datasets demonstrate that DockingApp RF’s performance is comparable to other state-of-the-art machine-learning- and deep-learning-based scoring functions. The new scoring function thus represents a significant advancement in terms of the reliability and effectiveness of docking compared to AutoDock Vina’s scoring function. At the same time, the characteristics that made DockingApp appealing to a wide range of users are retained in this new version and have been complemented with additional features.

## 1. Introduction

Bringing about a new therapeutic compound is an expensive and time-consuming process [1,2]. The number of drugs that progress through all the steps of a successful drug development process (from phase I clinical trials to drug approval) are very low: a recent study found that, out of 21,143 drug candidates, only 6.2% were able to reach the market [3]. In a drug discovery pipeline, a high binding affinity between the target protein and a small molecule is a crucial selection criterion. Binding affinity calculations can be carried out by means of experimental methods, including, but not limited to, isothermal titration calorimetry, electrophoresis and the fluorescence thermal shift assay. In order to cut both costs and time for the whole drug discovery process, several computational methods for binding affinity predictions have been developed [4,5,6,7]. Among them, molecular docking emerges for its efficiency and effectiveness. Molecular docking predicts the binding mode and affinity of a compound (sometimes in the form of a score related to it) for a target, allowing to prioritize top scoring molecules for further processing and subsequent testing. Molecular docking consists of a search algorithm, which finds the relative orientation of the ligand in the target binding site, and of a scoring function (SF), which predicts the binding strength between a given conformation of the ligand and the target. Currently, despite the fact that traditional docking tools show a good conformational space sampling ability, binding affinity predictions still have room for improvement [8]. Following the classification proposed by Bohm [9], SFs can be identified as force field-based, knowledge-based and empirical. Force field-based SFs compute the interaction energies of the protein–ligand complexes, with the nonbonded energy terms often referring to van der Waals and electrostatic terms only. Hydrogen bonding can be taken into account explicitly with an additional term or can be implicitly included into the electrostatic term. The solvation energy term, when included, is computed by continuum solvation models such as Poisson-Boltzmann (PB) and Generalized Born (GB) [10,11]. Examples of SFs include DOCK and AutoDock4 [12,13]. Knowledge-based SFs are based on the pairwise sum of statistical potentials between interacting atom pairs from protein–ligand complexes. In brief, the frequencies of a contact are correlated to its contribution to protein–ligand binding by applying an inverse Boltzmann analysis. Scoring functions of this type are implemented in tools such as PMF and DrugScore [14,15]. Empirical SFs compute the free energy of binding via the sum of several terms. These terms represent different energetic components of the protein–ligand binding (e.g., hydrogen bonding, lipophilic contacts, steric clashes, etc.), often relying on multivariate linear regression (MLR) or partial least-squares (PLS) to weigh each term [16]. Examples include AutoDock Vina and GOLD [17,18]. The last few years have seen the spread of nonlinear scoring functions; they are characterized by a nonrigid functional form, which is learnt from the data and capable of capturing complex relationships and hard-to-model features [19]. Some examples include, but are not limited to, KDEEP, a 3D convolutional neural network (CNN) based on a voxel representation of both proteins and ligands with their pharmacophoric properties [20], RF-Score, a random forest-based model using atomic contacts as features [21], AGL-SCORE, which encodes high-dimensional physicochemical properties into a graph-based representation of the ligand–protein complex [22], Onion-Net, a CNN based on element-specific contacts between proteins and ligands dependent on distance [23] and Pafnucy, a 3D CNN that employs some computer vision-derived strategies to encode the protein and the ligand [24]. Further details on scoring functions can be found in [16,19,25]. Compared to classical SFs, ML-based SFs result in an improved binding affinity prediction [26]. However, despite their performances and advantages, machine-learning-based SFs are not yet routinely used in docking simulations. This appears to be due, among other things, to the lack of a corresponding user-friendly software implementation.

This work presents a novel scoring function for the rescoring of molecular docking-predicted binding poses. The scoring function is implemented into a new version of DockingApp [27], named DockingApp RF, to permit the rescoring of AutoDock Vina [18] outputs with ease also for nonexpert users. The proposed function is based on a random forest model and a selection of specific features whose purpose is to make up for the existing limits of Vina’s original scoring mechanism.

This work provides a measure of the performance of the proposed scoring function through comparative tests with a number of competitors on the CASF-2013 and the CASF-2016 benchmarks [28,29].

## 2. Materials and Methods

The proposed scoring function, as said earlier, is based on a random forest model, which is a machine-learning method typically used for a variety of classification and regression tasks [30,31]. The following subsections describe the datasets used for training the model and the process employed for the selection of features to be included, as well as the tuning procedure.

### 2.1. Datasets

The data are retrieved from the PDBBind database, a collection of protein–ligand complexes with their experimentally determined binding affinity (expressed as Kd, Ki or IC_50_) derived from the Protein Data Bank (PDB) [32]. The PDBBind dataset consists of a general set containing all the protein–ligand structures in the database, a refined set and a core set. The refined set is a subset of high-quality complexes satisfying the strict criteria of selection, while the core set is derived from the refined set by clustering it according to BLAST [33]. The gold standard for testing novel SFs is the CASF benchmark, a test built upon the PDBBind core set and the refined set. The CASF benchmark offers different tests to assess the docking power, the scoring power, the ranking power and the screening power of a SF. This work is focused on the scoring power of the novel SF and, thus, on the ability to accurately predict a protein–ligand binding affinity. The proposed SF has been tested on both the CASF-2013 and CASF-2016 benchmarks. In order to comply with the benchmarks’ guidelines, it was trained on the 2013 and 2016 editions of the PDBBind refined set, respectively, after removing from the training set any complexes shared with the test set. Furthermore, by taking inspiration from the work of Boyles and coworkers [34], the effects on the binding affinity prediction caused by the size of the training set and the similarity between the latter and the test set have been investigated. For this purpose, several training sets have been derived from the PDBBind 2018 general set, and the models trained on them have been tested on a test set obtained from the union of the CASF-2016 and CASF-2017 test sets, henceforth dubbed the CASF-combined test set. The similarities can be identified at the ligand level and at the protein level. In order to investigate the effects of the first, the Tanimoto similarity of all ligand pairs has been calculated by means of RDKit [35]. Then, a novel training set was built by removing from the general set any structure whose ligand had a Tanimoto similarity 0.9 with any ligand in the test set. Regarding the protein similarity, different training sets were built by removing from the training set those complexes sharing a sequence identity above a certain threshold with those in the test set. Proteins were clustered by means of blastclust [33] with different identity values: 100%, 90%, 70%, 50% and 40%. The number of complexes in the training and test sets are reported in Table 1. Lastly, different ML models were tested, including support vector machine (SVM), linear regression (LR) and k-nearest neighbor (KNN).

### 2.2. Features Selection

In order to identify the physicochemical properties that determine the strength of the bond between a protein and its binders, three classes of features were extracted from the complexes:intermolecular contacts of the pharmacophoric types (phCo),variations of the solvent-accessible surface area upon binding (ΔSASA) andAutoDock Vina’s unweighted energy terms.

The details of these classes of features and the reasons behind their choices are described in the following subsections.

#### 2.2.1. Intermolecular Contacts

Intermolecular contacts distribution has shown a good predictive power in many scoring functions [24,36,37,38,39], due also to its ability to implicitly capture complex relationships and patterns, thus avoiding any arbitrary classification attempts. However, even in such a case, an arbitrary choice has to be made: the chemical representation of the atoms. To increase the density of the information, Ballester and coworkers opt for a simple representation by employing 9 different chemical elements only [40], also showing that a more precise chemical representation does not necessarily translate into a more accurate model. In this work, a customized version of the chemical representation of DOCK’s atom types was used, considering the good results reported in [41,42] using this representation. An additional chemical type identifying the presence of a metal ion (“MI”) was initially added to the nine standard pharmacophore types described by the “DOCK pharmacophore similarity score” but was later removed due to the fact that there was no real gain in the performance after its inclusion. This representation for assigning the pharmacophore type to a given atom takes into account the type of the atom and the types of the atoms bound to it. The pharmacophore types considered were positive (P), negative (N), donor-acceptor (DA), donor (D), acceptor (A), aromatic (AR), hydrophobic (H), polar (PL) and halogen (HA). Table 2 details the identification patterns of each pharmacophore type.

The intermolecular contacts were identified from the 3D protein–ligand complexes by using a KD-Tree algorithm applied to the distance, to account for both the short- and long-range interactions. The search zone in each iteration appears like a shell surrounding the atom under analysis, and the boundaries of each concentric shell are defined as:$\;i*d-d+\;d_{0}$ for the lower bound and $i*d+\;d_{0}$ for the upper bound, where i is the number of the level, d the spherical shell thickness (by default 2 Å) and $d_{0}$ an offset of 1 Å. By default, the search procedure was repeated for 9 different shells, sampling a total distance of 19 Å. The sampling procedure resulted in 900 contact features.

#### 2.2.2. Solvent Accessible Surface Area

The solvation effect is a driving force in molecular interactions, and its variations upon binding have been shown, in many cases, to be correlated with the binding affinity [43,44,45]. Nonetheless, few tools incorporate this effect into their scoring functions: as a matter of fact, most of the software tools for molecular docking lack any term accounting for the desolvation effect in their scoring function [46], because of the complexity involved in an accurate estimation of such an effect. Due to their inner capability of detecting complex relationships from simple features, machine-learning mechanisms are particularly suited to fill this gap in docking scoring functions. In this regard, the features selected for the proposed model are the variation of the solvent-accessible surface area (Δ*SASA*) upon binding for both the protein and the ligand and their contact surface area (*CSA*). Further details on these features can be found in the following Equations (1)–(3): (1)$$\Delta SASA_{protein}=SASA_{apo_{polar}}+SASA_{apo_{apolar}}-\;SASA_{holo_{polar}}-\;SASA_{holo_{apolar}}$$
(2)$$\Delta SASA_{ligand}=SASA_{ligand_{free}}-\;SASA_{ligand_{bound}}$$
(3)$$CSA=\;\frac{buriedSurface_{protein}+buriedSurface_{ligand}}{2}$$
where the polar and apolar *SASA* fractions for the *apo* (i.e., ligand-free) and *holo* (i.e., ligand-bound) proteins, together with the *SASA* of the ligand, were calculated by using Lee and Richards’ approximation implemented in the Python module of FreeSASA [47].

#### 2.2.3. Vina’s Energy Terms

In order to account for the physicochemical and steric complementarity between the ligand and protein and the entropic component, the unweighted energy terms derived from the scoring function of AutoDock Vina [18] and the number of rotatable bonds of the ligand were calculated. The energy terms used are two gaussian terms (Gauss 1 and Gauss 2), one repulsion term, one term taking into account the hydrogen bonds and one term for the hydrophobic interaction. The expressions of the energy terms used can be found in the Appendix A.

### 2.3. Performance Metrics and Errors Evaluation

Evaluation metrics employed to assess the performance of the model included Pearson’s correlation coefficient (*R*, Equation (4)), mean absolute error (*MAE*, Equation (5)), root mean squared error (*RMSE*, Equation (6)) and standard deviation (*SD*, Equations (7) and (8)), as calculated in CASF [28]:(4)$$R=\;\frac{\sum_{i=1}^{n}\left(x_{i}-\;\bar{x}\right)\left(y_{i}-\;\bar{y}\right)}{\sqrt{\sum_{i=1}^{n}\left(x_{i}-\;\bar{x}\right)}\;\sqrt{\sum_{i=1}^{n}\left(y_{i}-\;\bar{y}\right)}}$$
where n is the sample size and x and y the predicted and the expected pKd, respectively.
(5)$$MAE=\frac{1}{N}\overset{N}{\underset{i=1}{\sum }}\left|pK_{d_{pred_{i}}}-\;pK_{d_{exp_{i}}}\right|$$
(6)$$RMSE=\;\sqrt{\frac{1}{N}\sum_{i=1}^{N}\left(pK_{d_{pred_{i}}}-\;pK_{d_{exp_{i}}}\right)}$$
(7)$$SD=\;\sqrt{\frac{1}{N-1}\sum_{i=1}^{N}c^{2}}$$
(8)$$c=\;\left(\left(a*pK_{d_{pred_{i}}}+b\right)-\;pK_{d_{exp_{i}}}\right)$$
where a  is the slope and b the intercept of the linear regression line, while pred stands for predicted values and exp for the experimental values.

Along with the correlation coefficient, both the mean absolute error (*MAE*) and the root mean squared error (*RMSE*) were used for evaluating the errors of the prediction. The former (*MAE*) quantifies the magnitude of the errors in the prediction, and it is represented by the average of the sum of the absolute differences between the predicted values and the experimental values. The latter (*RMSE*) measures the relative deviations of the predicted values with respect to the experimentally determined values and is more sensitive to large errors with respect to the *MAE*. Furthermore, the standard deviation in the regression (*SD*) was used and calculated, as implemented in CASF [28].

## 3. Results and Implementation

To identify the best-performing machine-learning algorithm for affinity predictions, four different approaches were trained and tested using the features introduced above. In particular, the effects on the binding affinity prediction caused by the size of the training set and the similarity between the latter and the test set were assessed. Then, the informative value of each class of features and the ability of the algorithms to exploit them were investigated. The best-performing combination of algorithm and features was used in the CASF-2013 and CASF-2016 benchmarks in order to compare its effectiveness with a number of recent competing methods that either employ “classical” SFs or machine-learning-based SFs. Lastly, the docking power and the scoring power of the scoring function were assessed and directly compared with AutoDock Vina.

The details of these activities are reported in Section 3.1, Section 3.2, Section 3.3, Section 3.4 and Section 3.5, respectively, whereas Section 3.6 describes the implementation of the SF within DockingApp RF and the additional features introduced in the tool.

### 3.1. Model Comparison

In order to assess the best-performing approach, four different machine-learning algorithms were compared: support vector machine-based regression (SVM) (kernel = radiant basis function), linear regression (LR), random forest (RF) (n_estimator = 100) and k-nearest neighbor (KNN) (*k* = 10). SVM and LR were trained on normalized data. The models were trained on the data filtered based on the identity thresholds and tested on the combined test set. The results of the comparison are reported in Figure 1. SVM and RF are the best-performing models in each scenario, while KNN and LR yield the worst results. Removing similar ligands had less impact on the performance with respect to the elimination of similar proteins from the dataset. The greatest drop in performance was obviously observed with the first cut-off (by removing from the training set those structures having 100% protein sequence identity with those in the test set). These observations suggest that there is more redundancy in the proteins pool with respect to the ligand pool. Lowering the cut-off threshold further has a minor effect on the performance, with SVM slightly out-performing RF when taking into account structures with a maximum sequence identity of 70% or below.

### 3.2. Contribution of the Features

In order to identify the contribution of each class of features (Vina, pharmacophoric contact (phCo) and SASA), a model for each machine-learning algorithm was trained by using only one class of features or a combination of two of them for a total of 28 models. Each model was trained on the PDBBind2018 general set and tested on the CASF-combined test set. After the training, removing the nonzero variance resulted in 648 features for phCo, 6 features for Vina and 3 features for SASA. The results are reported in Figure 2, where standard Vina SF represents the baseline performance expressed as a Pearson’s correlation (R) (dotted line, R = 0.58). Models using SASA features alone yielded the worst performance, with a correlation coefficient on average comparable to the one of Vina’s SF. Conversely, the best performance was achieved using phCo features, with SVM and RF as the top performers. Interestingly, the refitted Vina energy terms showed a good correlation with the experimental data, underlining their informative value. Combining two classes of features increased the correlation with the experimental data on average; however, phCo emerged as the most informative class of features, being part of all the top-performing models and achieving alone a performance comparable to them. The SASA features seem to provide the least contribution when combined with another class of features. Using all the classes of features at the same time does not result in a significant increase in performance. Among the different models, those employing RF achieveed the best performance in every case.

In light of these results, three models were selected for further testing: RF-phCo, RF-All and RF-Vina+phCo. The tests involved a comparison of out-of-bag scores of the three models and a ten-fold cross-validation. The models were built by using 500 regression trees and max_features = 0.33, while the tests were performed on the PDBBind general set 2018 by using an 80:20 split to generate the training and test set. The best-performing model was RF-All, with a correlation of 0.75 and an out-of-bag score of 0.537; conversely, RF-phCo and RF-Vina+phCo achieved correlation coefficients of 0.74 and 0.75 and out-of-bag scores of 0.525 and 0.535, respectively. The ten-fold cross-validation, depicted in the Appendix A (Appendix A), offers a similar scenario, with RF-All and RF-Vina+phCo performing nearly identically, with a small edge for the first. Due to these results, the final model was built by employing all the classes of features (number of features = 657) and tested on the CASF benchmarks.

### 3.3. CASF-2013 Core Set Results

Using the CASF-2013 core set as a test set, both the models trained on the PDBBindv2013 refined set and the PDBBindv2018 general set were tested. In the first case, by reproducing the conditions of the CASF-2013 benchmark, the proposed model achieved a Pearson’s correlation coefficient of 0.77 and a SD of 1.41, with a RMSE of 1.55 and a MAE of 1.31. The performances of other methods on the same test set are reported in Table 3. Increasing the number of training samples produced an increase in performance, as the model trained on the PDBBindv2018 general set yielded a Pearson’s correlation coefficient of 0.79, a MAE of 1.23 and a RMSE of 1.49. A more detailed comparison with AutoDock Vina is reported in Figure 3. In the picture, it is possible to observe how AutoDock Vina achieves a correlation coefficient (R) of 0.57.

### 3.4. CASF-2016 Core Set Results

For the CASF-2016 core set, two different models were trained as well by using the PDBBindv2016 refined set and the PDBBindv2018 general set. In the CASF-2016 benchmark, the proposed scoring function achieved a Pearson’s correlation coefficient of 0.82 with a RMSE of 1.38 (SD = 1.26 and MAE = 1.13, Table 3), a performance comparable with KDEEP’s [20] (R = 0.82). Training the model on the general set resulted in an improvement of the correlation with a R of 0.83 (RMSE = 1.35 and MAE = 1.09). The direct comparison with AutoDock Vina is depicted in Figure 3; within such a test set, Vina achieved a R = 0.60.

### 3.5. Docking Power and Screening Power Testing

The Docking power was tested on both the 2013 and 2016 versions of the CASF benchmark, while the screening power was tested on a subset of the DEKOIS2.0 dataset. In the CASF benchmark, the docking power is assessed starting from a distribution of different ligands conformations around the binding site. The SF is called to recognize those ligands with a RMSD within 2 Å from the native one. The performance is described by three values, which account for the number of times the SF places a native-like ligand conformation in the top one, top two or top three predictions. In both tests, the scoring function stands at the bottom of the ranking reported in Figure 5 of the paper by Su et al. [29], with a score of 27.7%/38.2%/46.6% and 35.3%/50.4%/57.7% in the CASF-2013 and CASF-2016 benchmarks, respectively. The ability to distinguish true binders from decoy compounds was tested on the DEKOIS 2.0 set. From the latter, nine complexes were selected: four from the challenging category (VEGFR2, EGFR, DHFR and FXA), three from the moderately challenging category (A2A, CYP2A6 and P38a) and two from the less-challenging category (PDE4b and PRKCQ). For each target, docking simulations for all of the true binders and decoys available were performed, the area under the curve (AUC) score calculated and the ROC curve plotted. Vina’s SF slightly outperformed the proposed SF in three cases out of nine (P38a, PDE4b and PRKCQ) while showing a comparable performance in three cases out of nine (DHFR, FXAA and A2A). In the last three cases (EGFR, VEGFR2 and CYP2A6), DockingApp RF performed significantly worse than Vina. All of the ROC curves and the AUC values are available in the Appendix A (Appendix A).

### 3.6. DockingApp RF’s Implementation and New Features

The scoring function discussed so far was implemented in DockingApp RF, a desktop application for docking and virtual screening that is meant as a user-friendly interface to AutoDock Vina, taking over from the earlier DockingApp. The new scoring function is thus used both for the Docking and Virtual Screening features of DockingApp RF, as well as for the newly introduced Replicated Docking described below.

#### 3.6.1. Replicated Docking

A useful new feature introduced in DockingApp RF is the so-called “Replicated Docking”. This is an additional operation that has been made available alongside the original Docking and Virtual Screening modes and allows the user to perform a molecular docking against a given target to be repeated a user-defined number of times, each with a different random seed. The results of the Replicated Docking show the best poses collected from each repetition. The underlying idea motivating the inclusion of this new feature lies in providing users with the possibility of further exploring the conformational space of the poses, all the while identifying the most consistent conformation observed among a significantly high number of executions.

#### 3.6.2. Extension of the Drug Library Collection

Due to the relevance and success achieved by drug repurposing initiatives [49,50,51], the ready-to-be-docked library of the original DockingApp is now expanded in DockingApp RF. Indeed, the original collection of about 1400 FDA-approved drugs [52] is integrated with 4288 drugs approved by major national regulatory agencies and includes tautomers and different protonation states. The structures were downloaded from ZINC [53] in a mol2 format and then converted to .pdbqt by using the prepare_ligand4.py script from AutoDock Tools.

#### 3.6.3. DockingApp RF’s Additional Functionalities

Further refinements in terms of user experience have been introduced as well in DockingApp RF. Specifically, the tool now enables users to recall earlier executions from a convenient drop-down menu for each of the core operations, so that they can be easily launched again without setting the corresponding parameters anew or quickly adjusted according to the users’ specific needs.

#### 3.6.4. Technology, Requirements, Availability and Execution Times

DockingApp RF’s mechanism for computing the proposed novel scoring function was developed in Python. The pharmacophoric types are assigned by using Open Babel 2.4 and Pybel [54], while the range of contacts is identified through a KD-Tree algorithm implemented in the scipy.spatial module. SASA features are calculated through the FreeSASA library [47]. The random forest model is implemented by using the scikit-learn library v0.20 with the scikit-learn.ensemble module. The randomized search is performed via the sklearn.model_selection module. DockingApp RF’s main application, just like its predecessor, is developed in Java SE and is freely installable on a number of operating systems, including Windows 7/Vista/8/10 and Unix; the corresponding distribution packages of the software can be found at http://www.computationalbiology.it/software.html. A version for Mac OSX will be made available in the near future. In order to run it, it requires the following additional software and libraries: Java 8, Python 3.6+, FreeSASA 2.x, Openbabel 2.4+ and Sklearn 0.2; the installation of these software packages and libraries is automated during the DockingApp RF’s installation procedure by means of a dedicated script. More info can be found in DockingApp RF’s installation instructions. Execution times for docking and rescoring have been measured on a number of different hardware configurations. On a server sporting a dual-CPU configuration with two Intel Xeon E5-2640 v4 CPUs, 64GB RAM and SSD storage, with 10 cores reserved for DockingApp RF, screening a protein against 1466 compounds took about 30 h, which resulted in an average time of 77 s per compound. On a desktop configuration with an Intel I7 7700 CPU, 32GB RAM and SSD storage, with six cores reserved for DockingApp RF, it took slightly more than 75 h for the same task, which is about 3.5 min per compound.

## 4. Discussion

DockingApp RF’s new scoring function lies upon one core intuition: the contacts between protein and ligand, enforced by an accurate chemical representation, can be turned into features to be fed to a machine-learning algorithm, in accordance to what was already observed and performed by Ballester and coworkers [36]. Starting from this observation, the present work aimed to improve the predictive power of the scoring function of AutoDock Vina by providing a user-friendly and ready-to-use machine-learning-based scoring function. In the latest years, a number of other, successful scoring functions have been developed, but too often, they are published in the form of obscure code or difficult-to-use packages. In the present work, instead, such a scoring function was implemented within an intuitive tool, building up from the results of the earlier DockingApp and now providing not only an AutoDock Vina interface but, also, the newly developed scoring function’s binding affinity values, along with Vina’s original ranking (Figure 4).

Among the various ML algorithms tested in this work, random forest showed its advantage with respect to the others, being able to better exploit the increased number of features. A similar behavior was observed for SVM but, conversely, not for KNN and LR, whose performances are far behind RF and SVM. Speaking of features, among the three kinds of those employed in this work, phCo was the one yielding the best performance, either taken by itself or in addition to another class. This could be due to the higher number of elements that make it up (648 features for phCo against three features for SASA and six for Vina). The great number of features can also help explain the diluted features’ importance observed in the final model (Appendix A). Comparing the performance of the models in the ten-fold cross-validation test, it appears that adding Vina energy terms results in a marked gain in performance, while the addition of SASA terms is characterized by a marginal improvement. However, by looking at the contribution of each single feature in the final model, the SASA features seem to contribute significantly (Appendix A). It is not clear whether the poor performance of SASA features could be related to their naive implementation; in order to better clarify this point, further studies need to be carried out by comparing various methods to calculate the solvent accessible surface area and their corresponding implementation. The final model tested in both CASF-2013 and CASF-2016 consisted of 657 features and 500 regression trees and was set to 0.33. The performance achieved in the CASF benchmarks confirmed the potential of machine-learning approaches in binding affinity predictions and put the present on-par with other state-of-the-art methods. However, preliminary studies carried out to assess the performance of the proposed method in correctly assessing the native conformation of a ligand and in distinguishing true binders from a decoy compound revealed that it has yet to be optimized for such a task. Indeed, in both tasks, the performance of DockingApp RF fell behind AutoDock Vina’s. The suboptimal performance in terms of docking power and screening power is often a physiological issue in machine-learning-based scoring functions focused on the prediction of binding affinities. A jack-of-all-trades machine-learning-based scoring function is still a chimera, and as a consequence, task-specific scoring functions are usually developed, as demonstrated in [38] and [55]. Moreover, a recent study [26] shed a gloomy light on machine-learning-based scoring functions used for virtual screening purposes, due to their limited levels of performance on those targets that are dissimilar to the proteins in the corresponding training sets. By looking at the range of values predicted by the proposed SF, it is possible to observe how it is narrower than AutoDock Vina’s. This is a recurrent observation for machine learning: these methods tend to underestimate the affinity of tight binders and overestimate the affinity of loose binders. By looking at the distribution of the affinity values in the training sets, it is possible to better understand the basis of this observation. Indeed, as it is reported in the Appendix A (Appendix A), the majority of the complexes in the training sets sport a pKd between 3 and 9; this effect is greater in the case of tight binders and lower in the case of loose binders.

Lastly, it is worth dwelling on the performance of the methods based on deep learning. Despite the potential of these approaches, their implementation in binding affinity prediction activities is still lagging behind in terms of sheer performance, compared to simpler methodologies such as machine learning. As for the reasons for this, it seems to be an issue related to the size of the training set when a specific type of architecture is employed. It is, perhaps, the case of DeepBindRG [56]. As a matter of fact, this method makes use of the ResNet convolutional neural network (CNN) architecture [57], a model largely used in object detection, image recognition and computer vision in general, which ranked first in the ILSVRC 2015 classification task; for these tasks, the neural network can rely on databases like ImageNet, which currently contains more than 14,000,000 labeled images [58]. On the other hand, the current gold standard for protein–ligand binding affinity prediction, the PDBBind, contains little more than 15,000 complexes with binding affinity data. It seems that this architecture is particularly hungry in terms of the size of the training sample, and the performance of OnionNet [23] (Table 3), which does not use a protein–ligand representation borrowed from the picture depiction science, seems to confirm this observation.

## 5. Future Development

Given the issues mentioned throughout the discussion, the gain in accuracy for the binding affinity prediction task that could be obtained thanks to a more precise chemical description of the contact between a protein and a ligand is an issue of which the authors of the present work are well aware. For this reason, while currently being distributed with a ready-to use model, future releases of DockingApp RF plan to include a built-in tool for allowing more advanced users to train a custom model on a given training set of protein–ligand complexes. In this way, a scoring function more tailored to a specific need could then be employed within the application accordingly.

Besides, from a technological standpoint, both DockingApp RF and the original DockingApp [27] are also planned to be released in the near future as a single server-side web application in a similar fashion to how the authors’ earlier released tools for catalytic site detection and binding site detection, ASSIST [59] and LIBRA [60], respectively, were merged and included back then in the LIBRA-WA [61] web application. This would, on the one hand, provide users with a convenient way of accessing DockingApp RF’s functionalities regardless of the specific platform or operating system while leveraging the computational power of a dedicated server for a more efficient computation of the docking results with respect to a desktop computer. On the other hand, this would also pave the way to a subsequent integration of DockingApp RF’s procedure within the LIBRA-WA ecosystem itself, with the ultimate purpose of bringing about a comprehensive and high-performing tool for the whole life cycle of in silico drug design.

## 6. Conclusions

The accurate prediction of the binding affinity between a small molecule and a protein is still an issue for molecular docking software, as well as a bottleneck in structure-based drug discovery and design, despite the results achieved so far in the conformational search. For these reasons, this work presented a state-of-the-art scoring function for molecular docking based on a random forest algorithm that exploited the interatomic distance of different protein–ligand atom types, Vina’s energy terms and the variation of the solvent-accessible surface area upon binding. The results on the tests carried out on both CASF-2013 and CASF-2016 benchmarks demonstrated an excellent performance with respect to other machine-learning- and deep-learning-based scoring functions. As it stands, this novel scoring function is thus proposed as a direct improvement of the binding affinity prediction ability of AutoDock Vina and was implemented in DockingApp RF, a desktop application taking over from DockingApp, a tool initially meant as a user-friendly interface to AutoDock Vina itself. As a result, DockingApp RF now sports a more accurate prediction of ligands’ binding affinity thanks to the seamless integration of the novel scoring function while retaining all of the user-friendly characteristics that made DockingApp successful.

## Figures and Tables

**Figure 1 ijms-21-09548-f001:**
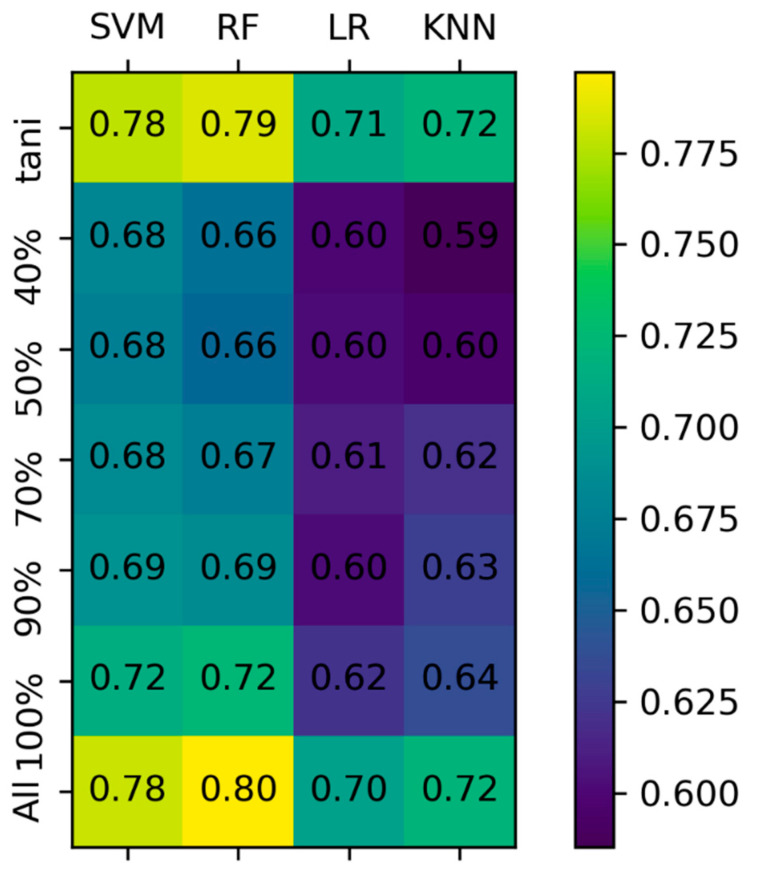
Pearson correlation coefficient of predicted versus experimental binding affinity achieved by different machine-learning models when trained on filtered datasets and tested on the combined test set. The training sets are derived from the PDBBindv2018 general set by applying a filter for sequence identity and ligand similarity. Support vector machine-based regression (SVM), linear regression (LR), random forest (RF) and k-nearest neighbor (KNN).

**Figure 2 ijms-21-09548-f002:**
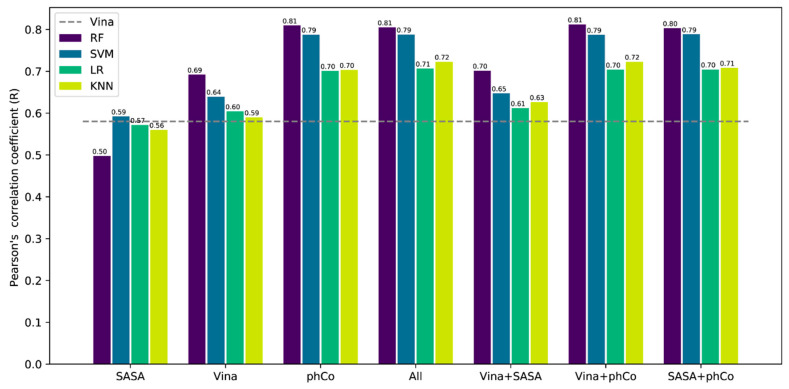
Comparison of the contribution of each class of features in the various machine-learning models (SVM: support vector machine, LR: linear regression, RF: random forest and KNN: k-nearest neighbor). SASA refers to a solvent-accessible surface area, Vina refers to AutoDock Vina’s energy terms and phCo refers to intermolecular contacts. The dotted line indicates the baseline performance represented by AutoDock Vina.

**Figure 3 ijms-21-09548-f003:**
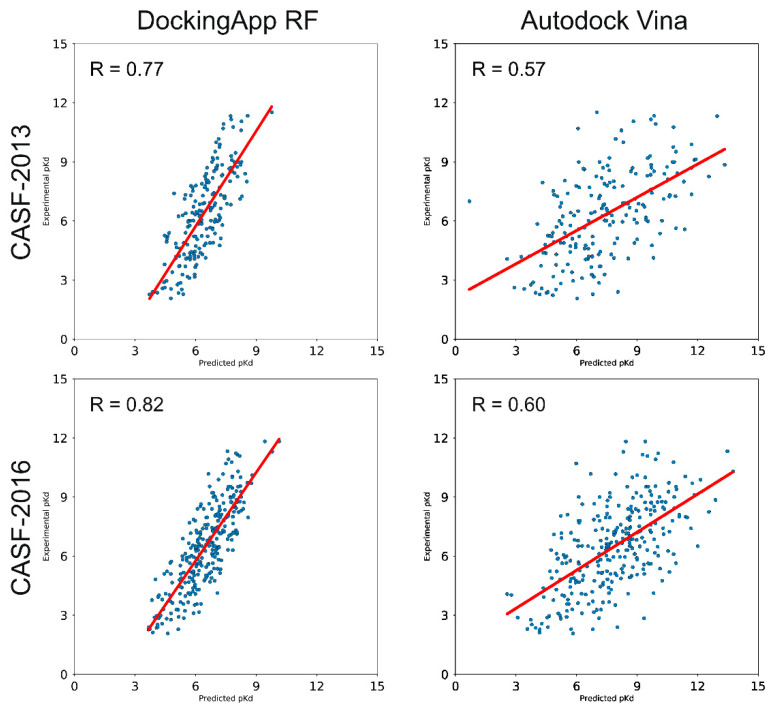
Performance comparison between DockingApp RF’s and AutoDock Vina’s scoring functions on the CASF-2013 and CASF-2016 benchmarks.

**Figure 4 ijms-21-09548-f004:**
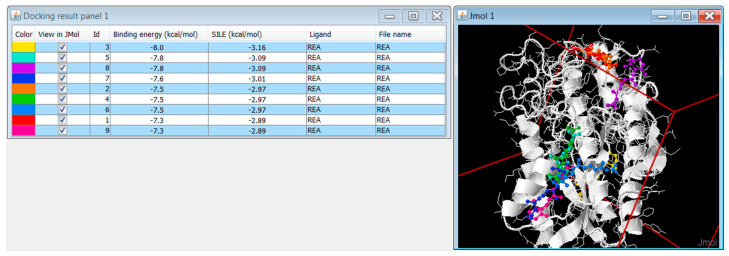
Screenshot of DockingApp RF’s output window, displaying the predicted binding energy values of the novel scoring function, along with the SILE (Size-Independent Ligand Efficiency [27]) values.

**Table 1 ijms-21-09548-t001:** Summary table of the training and test sets used. CASF-combined refers to a set obtained from the union of the complexes in CASF-2013 and CASF-2016. The trailing number of the training sets refers to the identity percentage used for filtering out similar complexes between the test and training set, whereas the -Tani suffix refers to the set where complexes were removed via the Tanimoto coefficient, as described above.

Benchmark	Training-Set	*n*. Complexes	Test Set	*n*. Complexes
CASF-2013	PDBBindv2013 refined set	2764	PDBBindv2013 core set	195
CASF-2016	PDBBindv2016 refined set	3772	PDBBindv2016 core set	285
CASF-combined	PDBBindv2018 general set 100	12,002	PDBBindv2013 core set + PDBBindv2016 core set	370
CASF-combined	PDBBindv2018 general set 90	10,943	PDBBindv2013 core set + PDBBindv2016 core set	370
CASF-combined	PDBBindv2018 general set 70	10,523	PDBBindv2013 core set + PDBBindv2016 core set	370
CASF-combined	PDBBindv2018 general set 50	10,173	PDBBindv2013 core set + PDBBindv2016 core set	370
CASF-combined	PDBBindv2018 general set 40	9597	PDBBindv2013 core set + PDBBindv2016 core set	370
CASF-combined	PDBBindv2018general set Tani	13,194	PDBBindv2013 core set + PDBBindv2016 core set	370

**Table 2 ijms-21-09548-t002:** Pharmacophore-type assignment equivalence. Parentheses and brackets indicate bond information: “( )” means atoms that must be bonded to the parent atom, while “[ ]” specifies atoms that must not be bonded. * indicates any atom.

Pharmacophore Type	SYBYL Atom Type
P = Positive	N.4 (4 *)
	N.2 (3 *)
	N.pl3 (C.cat)
N = Negative	O (C (2 O or S [*]))
	O (P (2 O or S [*]))
	O (S (3 O [*]))
	S (C (4 *)) [*]
	S (C (2 (O or S [*])))
DA = Donor-acceptor	O (H)
	N.3 (H)
	N.2 (H)
	N.pl3 (H)
	S (H)
D = Donor	N.ar (H)
	N.am (H)
A = Acceptor	O Default
	N.3
	N.1
	N.ar (2 *)
	N.pl3
	S [3 *]
AR = Aromatic	N.ar
	C.ar
H = Hydrophobic	C [N] [O] [F] [P] [S]
PL = Polar	N.am
	S (3 *)
	C (N) (O) (F) (P) (S)
	P
HA = Halogen	F
	Cl
	Br
	I

**Table 3 ijms-21-09548-t003:** Performance comparison of different scoring functions on CASF-2013 and CASF-2016. Scoring functions marked with * used as a training set the PDBBind general set. R: Pearson’s correlation coefficient, SD: standard deviation, RMSE: root mean squared error and MAE: mean absolute error. SF: scoring function.

**CASF-2013**				
**SF**	**R**	**SD**	**RMSE**	**MAE**
*AGL-SCORE*	0.79	n.a.	1.97	n.a.
*DockingApp RF **	0.79	1.26	1.38	1.13
*OnionNet **	0.78	1.45	1.50	1.21
*DockingApp RF*	0.77	1.41	1.55	1.31
*RF-Score-v2*	0.74	1.50	1.60	n.a.
*RF-Score-v3*	0.74	1.50	1.59	n.a.
*Pafnucy **	0.70	1.61	1.62	1.51
Δ*VinaRF20*	0.69	1.64	n.a.	n.a.
*DeepBindRG*	0.64	1.73	1.82	1.48
*X-Score [48]*	0.61	1.78	n.a.	n.a.
*AutoDock Vina*	0.57	n.a.	2.4	1.95
**CASF-2016**				
**SF**	**R**	**SD**	**RMSE**	**MAE**
*AGL-SCORE*	0.83	n.a.	1.73	n.a.
*DockingApp RF **	0.83	1.26	1.38	1.13
*DockingApp RF*	0.82	1.26	1.38	1.13
*KDeep*	0.82	n.a.	1.27	n.a.
*OnionNet **	0.82	1.26	1.28	0.98
*RF-Score-v2*	0.81	1.28	1.42	n.a.
*RF-Score-v3*	0.80	n.a.	1.39	n.a.
*Pafnucy **	0.78	1.37	1.42	1.13
Δ*VinaXGB **	0.80	1.32	n.a.	n.a.
Δ*VinaRF20 **	0.73	1.26	n.a.	n.a.
*X-Score [48]*	0.63	1.69	n.a.	n.a.
*AutoDock Vina*	0.60	n.a.	2.35	1.94

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
