# Peer review of "DockingApp RF: A State-of-the-Art Novel Scoring Function for Molecular Docking in a User-Friendly Interface to AutoDock Vina"

_ijms, 2020, doi:10.3390/ijms21249548_

Round 1

Reviewer 1 Report

This is an interesting manuscript investigating on DockingApp RF, a scoring function for molecular docking method. This new method demonstrate good performance compared to other algorithm, and make the DockingApp RF a user-friendly graphical interface for AutoDock Vina. This is a well-organized manuscript, with good presentation, clear workflow and in-depth discussion. There are some typo error need to be correct but I still recommend publish this manuscript after minor revision.

Author Response

We thank very much the Reviewer for careful reading of the manuscript and appreciation of our work. We carefully revised the text and corrected a few typos.

Reviewer 2 Report

The authors implemented a scoring function to predict binding affinities and integrated this function into an application named DockingAppRF. The authors employed multiple benchmarks for the proposed function and assessed its performance against published methods. The study is solid and of importance for drug design. However, there are a few points to be addressed.

  1. Table 1. Add an additional column telling where the test sets were derived (like what you did for the second column)
  2. While you select and described three major classes of features, you probably included hundreds or even thousands of features for building the models. Did you perform feature selections based on how each feature affected the performance of the function? How many features did you use in your final model?
  3. How did you determine the K for KNN and n_estimator for RF? Have you performed preliminary evaluations to choose an optimal parameter for those methods?

Author Response

First of all we wish to thank the Reviewer for his/her useful comments. Here follows a point-to-point-reply:

  1. A test set column specifying the test set used in each benchmark has been added to Table 1.
  2. The number of features in the model is reported in line 266. Besides removing zero variance features, no further features selection has been performed. However, the ten most important features have been reported in the Supplementary Materials Fig. S5 and Fig. S6.

In order to determine the K for KNN, integer values of K from 1 to 20 were tested and the one yielding the best performance (i.e., K = 10) was used to obtain the results reported in Figure 1 of the main text. The value of the n_estimator for RF resulted from the outcome of an optimization procedure of the hyperparameters. We performed a grid search looking for the best combination of hyperparameters. The resulting choice of 500 as the number of estimators was further corroborated by the analysis of other works in the literature on the same subject.